# miR-7-5p and Importin-7 Regulate the p53 Dynamics and Stability in Malignant and Benign Thyroid Cells

**DOI:** 10.3390/ijms26125813

**Published:** 2025-06-17

**Authors:** Abeer Al-Abdallah, Iman Jahanbani, Bashayer Al-Shammari

**Affiliations:** 1Pathology Department, College of Medicine, Kuwait University, P.O. Box 24923, Safat 13110, Kuwait; 2Research Core Facility, Health Sciences Center, Kuwait University, P.O. Box 24923, Safat 13110, Kuwait; iman.jahanbani@ku.edu.kw (I.J.); bashayr.alshimari@ku.edu.kw (B.A.-S.)

**Keywords:** PTC, miR-7-5p, importin 7, p53, MDM2

## Abstract

Thyroid carcinogenesis has multiple hallmarks, including evasion of tumor suppressors. Reactivation of wild-type p53 function is the ultimate goal in cancer therapy, which requires an understanding of the p53 suppression mechanism specific to the cancer type. MiR-7-5p and IPO7 are implicated in the pathogenesis of several human diseases. This work aims to investigate the role of miR-7-5p and IPO7 in p53 regulation in papillary thyroid cancer (PTC) cells. Primary cultured thyroid cells and FFPE thyroid tissues from PTC and benign cases were used. Functional experiments were performed by transfection with IPO7 siRNA or miR-7-5p mimic/inhibitor, followed by apoptosis and luciferase reporter assays, immunoblot assays, and RT-PCR. The expression and subcellular localization of IPO7, p53, MDM2, and ribosomal proteins (RPL11 and RPL5) were studied by immunofluorescence staining and confocal microscopy. The results show that IPO7 is overexpressed in PTC and regulated by miR-7-5p. Modulation of IPO7 expression in cultured thyroid cells altered the nucleocytoplasmic shuttling of p53, MDM2, RPL11, and RPL5, in addition to the p53 protein level and activity. The expression pattern of IPO7, p53, and MDM2 in cultured thyroid cells and clinical thyroid tissue specimens confirmed the association between IPO7 overexpression and reduced p53 stability in PTC. In conclusion, the data here show that p53 level and activity are differentially controlled in malignant and benign thyroid cells through miR-7-5P/IPO7-mediated regulation of RP-MDM2-p53 nucleocytoplasmic trafficking. In PTC, downregulation of miR-7-5p with consequent overexpression of IPO7 might be a protective mechanism used by cancer cells to evade p53 growth suppression during carcinogenesis.

## 1. Introduction

Thyroid cancer is the most common type of endocrine malignancy, and its incidence has increased worldwide over the last decade [1,2]. Papillary thyroid cancer (PTC) is the most common type of thyroid cancer, accounting for approximately 80% of all cases [3]. Although PTC is considered a nonlethal and treatable cancer, it can develop into aggressive variants and metastasize to neck lymph nodes [4,5]. PTC development is associated with multiple sporadic genetic alterations affecting mainly the mitogen-activated protein kinase (MAPK) and phosphatidylinositol 3-kinases (PI3K) pathways. Gene abnormalities such as mutations in *BRAF, RAS, TERT, TP53*, and *RET/PTC* are biomarkers used for classification and diagnostic purposes [5,6]. During transformation and progression, thyroid cancer cells undergo multiple genetic and epigenetic changes, including suppression of p53 activity [6,7]. Studies showed that *TP53* mutation is common in undifferentiated thyroid tumors (50–80%); however, it is less frequent in PTC (3–11%) [8,9]. Therefore, inactivation of the wild-type p53 by other mechanisms is presumed to contribute to PTC development [10]. In our previous work, we demonstrated that miR-7-5p is significantly downregulated in papillary thyroid cancer and regulates the expression of epidermal growth factor receptor (EGFR) in thyroid neoplasms [11,12]. In parallel with its downregulation, miR-7 appears to function as a tumor suppressor in thyroid and other cancers and to indirectly promote the p53 signaling pathway [13,14,15,16,17]. On the other hand, miR-7-5p was found to activate oncogenes and suppress the p53-dependent apoptotic pathway [18,19].

Importins are molecules that mediate the nuclear translocation of many proteins and regulate many biological functions, including proliferation, differentiation, and cell death [20,21,22]. IPO7 belongs to the importin-β family and is a transporter of many signaling proteins and transcription factors important in tumorigenesis, such as HIF1-α [23], c-Jun [24], and SMADs [25]. IPO7 is implicated in several human malignancies by facilitating the nuclear import of oncogenes to support tumor cell growth, migration, and metastasis [26,27,28,29,30,31]. Few studies have linked IPO7 to p53 regulation [32,33]. A recent study performed in pancreatic cancer tissues showed that the overexpression of IPO7 facilitated the malignant phenotypes of pancreatic cancer cells through an IPO7 and p53 positive feedback loop [33]. There are no previous reports on IPO7 expression, its regulation, or its contribution to the pathogenesis of PTC. This work aims to investigate the role of miR-7-5p and IPO7 in regulating the p53 activity in PTC.

## 2. Results

### 2.1. Expression of IPO7 and p53 in PTC Clinical Samples

IPO7 mRNA expression is significantly upregulated in PTC (4608 folds, *p* < 0.0001) and NIFTP (290 folds, *p* < 0.0001) compared to FND (Table 1). The high IPO7 expression is associated with a significant downregulation of miR-7-5p in PTC (−1573 folds, *p* < 0.0001) and NIFTP (−630 folds, *p* = 0.0003) (Table 1). Immunofluorescence staining and in situ hybridization experiments confirmed the high expression of the IPO7 protein and the low expression of miR-7-5p in PTC tissue samples, while in FND, there was a low expression of IPO7 and high expression of miR-7-5p (Figure 1A). The *TP53* mRNA expression showed no significant difference in PTC (−1.3 folds) and NIFTP (1.01 folds) compared to FND (Appendix A). However, immunofluorescence staining showed that p53 is not expressed in tumor cells in all PTC samples tested, while in FND, p53 was evident in the nuclei and cytoplasm of follicular cells (Figure 1B). MDM2 appeared in the nuclei and cytoplasm of follicular cells in PTC and FND tissues (Figure 1B). Additional immunohistochemical staining confirmed the lack of p53 expression and the high expression of IPO7 in PTC samples (Appendix A).

### 2.2. p53 Activity Is Suppressed in PTC and Is Rescued by miR-7-5p Gain of Function

Primary cultured thyroid cells from PTC and FND samples were prepared for functional studies. Luciferase assay showed that the endogenous activity of p53 was significantly lower in PTC (mean change = −2.89 folds, *p* = 0.01) compared to FND (Table 2). Reduced activity of pRb and Myc pathways was also significantly lower in PTC compared to FND (pRb mean change = −2.3 folds, *p* = 0.02) (Myc mean change = −3.04 folds, *p* = 0.01) (Table 2). miR-7-5p gain of function resulted in a significant reduction in IPO7 expression in all the five PTC primary cell cultures with *p* < 0.002 (Table 3). Luciferase assay showed that miR-7-5p gain of function restored the activity of the p53 and Myc pathways in four out of the five PTC primary cell cultures with *p* = 0.04 and *p*= 0.01, respectively (Table 3). Effects on other cell death pathways were not statistically significant. Apoptosis and viability assays showed that miR-7-5p gain of function increased the number of dead/apoptotic cells and reduced the viability of PTC cultured cells (Appendix A).

### 2.3. IPO7 Partial Depletion Increased the Availability of p53 Protein and the Nuclear Colocalization of MDM2 with the Ribosomal Proteins RPL11 and RPL5 in Primary Cultured PTC Cells

To investigate the effect of IPO7 on p53, primary cultured PTC cells were transfected with IPO7 siRNA. Partial depletion of IPO7 expression was confirmed by RT-PCR and immunofluorescence staining (Appendix A). Partial depletion of IPO7 resulted in a significant increase in p53 protein (*p* = 0.0026) (Figure 2A) and in the accumulation of p53 and MDM2 in the nuclei of the transfected PTC cells (Figure 2B). Partial IPO7 depletion also significantly increased the expression of miR-7-5p (Appendix A). Investigation of the involvement of ribosomal proteins by immunofluorescence revealed that RPL11 strongly accumulated in the nuclei of cultured PTC cells transfected with IPO7 siRNA, while it predominantly localized to the cytoplasm in the control cells (Figure 3A). RPL11 colocalized with MDM2 in the cell nuclei after IPO7 depletion (Figure 3A). RPL5 showed a similar pattern of nuclear accumulation and colocalization with MDM2 in the treated cells (Figure 3B).

### 2.4. p53 Regulation by miR-7-5p/IPO7 in the Normal Thyroid Cell Line Nthy-Ori 3-1

Nthy-ori 3-1 is a normal thyroid cell line that we used to confirm the regulation of p53 by miR-7-5p/IPO7. Nthy-ori 3-1 cells express miR-7-5p and a low level of IPO7 (Figure 4A). To recapitulate the expression profile in PTC, loss of function experiments using miR-7-5p inhibitor were performed. Inhibition of miR-7-5p increased the expression of IPO7 (Figure 4A, Appendix A). The level of p53 protein expression in Nthy-ori 3-1 cells is reduced after miR-7 inhibition and is rescued with IPO7 partial depletion (Appendix A). Nthy-ori 3-1 cells showed a shift of p53 and MDM2 from the nuclei to the cytoplasm after miR-7-5p inhibition (Figure 4B). RPL11 and RPL5 were found to be localized with MDM2 in the nuclei of untreated Nthy-ori 3-1 cells (Figure 4C,D). Upon inhibition of miR-7-5p, MDM2 and RPL11 appeared to be more focused in the cytoplasm, while RPL5 remained to be strongly detected in the nuclei and cytoplasm of the treated cells (Figure 4C,D).

## 3. Discussion

The new WHO classification divided thyroid tumors into benign, low-risk, and malignant neoplasms based on pathologic features, molecular characteristics, and biological behavior (36). Classic PTC is classified as a high-risk subtype of malignant follicular cell–derived neoplasm. Noninvasive follicular thyroid neoplasms with papillary-like nuclear features (NIFTPs) were classified as low-risk, follicular, cell-derived neoplasms. Multifocal hyperplastic lesions are now referred to as “thyroid follicular nodular disease (FND)” and are classified as benign conditions [34]. Many genes with diverse cellular functions have been shown to contribute to the PTC phenotype and functional characteristics [4,5,6,7,10]. This is the first report on IPO7 expression and its potential contribution to PTC pathogenesis. Here, we demonstrated that the IPO7 gene is overexpressed in PTC and NIFTP compared to FND (Table 1). In our samples, no correlation of IPO7 expression with any of the aggressive tumor characteristics were detected. The IPO7 protein appeared to be highly expressed in the cytoplasm and nuclei of tumor cells (Figure 1A). These results agree with the profile reported in the Protein Atlas database [Human Protein Atlas]. IPO7 expression inversely correlated with miR-7-5p expression and could be reversed with miR-7-5p gain of function (Table 1, Table 3). This profile of IPO7 and miR-7-5p regulation, which is consistent across high-risk (PTC) and low-risk (NIFTP) neoplasms, suggests that upregulation of IPO7 and downregulation of miR-7-5p are involved in neoplastic transformation rather than in the progression of thyroid tumors.

The altered function of wild-type p53 has been described in thyroid cancer and has been attributed to multiple mechanisms affecting p53 transcriptional activity, protein stability, or downstream signaling [35,36]. Our results showed reduced activity of p53 and other signaling pathways involved in cell death and apoptosis in PTC compared to FND (Table 2). This reduced p53 activity in PTC is not caused by *TP53* gene downregulation as indicated by the lack of significant difference in *TP53* gene expression between PTC and FND (Appendix A). It is well established that MDM2 negatively regulates p53 by binding to its N-terminal domain and blocking its access to the transcription machinery or by targeting it for proteasomal degradation. During these regulatory activities, MDM2 and p53 adopt diverse intracellular localizations that have functional consequences. MDM2 export to the cytoplasm is required for the degradation of p53, while inhibition of this export modifies the ability of MDM2 to block p53 and leads to p53 stabilization and activation [37,38,39,40,41]. Our results show that in PTC, MDM2 is localized to the cytoplasm of tumor cells with no evidence of p53 protein presence. In FND, both p53 and MDM2 are observed in the cytoplasm and nuclei of follicular cells (Figure 1B). This p53/MDM2 expression pattern in our samples agreed with that published in the Human Protein Atlas. Reduced nuclear expression of p53 has been previously reported in cancers such as melanoma, breast, and colorectal cancers [42,43]. In PTC, one study reported that nuclear accumulation of the p53 protein is only associated with the dedifferentiation of papillary carcinoma [44]. On the other hand, our experiments demonstrated that miR-7-5p gain of function restored the activity of the p53 pathway in PTC cells (Table 3). Partial depletion of IPO7 in PTC cells significantly increased the level of p53 protein and resulted in the accumulation of p53 and MDM2 in the cells’ nuclei (Figure 2A,B). Moreover, partial IPO7 depletion significantly increased the expression of miR-7-5p, which can be a feedback mechanism consistent with the increased p53 activity seen with miR-7-5p gain of function (Appendix A, Table 3). Altogether these results indicate that miR-7-5p and IPO7 modulate the activity of p53 through controlling its dynamics and stability

Carcinogenesis is usually associated with an increase in ribosome biogenesis, which is believed to be one of the consequences of the tumor suppressors’ altered functions [45,46]. Loss of function of p53 not only results in loss of cell proliferation control but also in upregulation of ribosomal genes [9,47,48,49]. Disturbance of ribosomal biogenesis and nuclear transport triggers the accumulation of ribosomal proteins (RPs), such as RPL5 and RPL11, in the nucleoplasm. The binding of RPL5 and RPL11 to MDM2 and the inhibition of its ubiquitin ligase activity have been suggested to be critical steps in p53 activation in response to cellular stress. RPL5 and RPL11 bind and sequester MDM2 in the nucleoplasm, resulting in p53 stabilization and activation and consequent proliferation arrest [50,51,52,53]. Cancer cells use multiple methods to evade this mechanism and continue to grow [54,55]. IPO7 has been linked to the ribosomal biogenesis stress response in one study that showed that partial IPO7 depletion resulted in RPL5/RPL11-mediated inhibition of MDM2 and p53 nuclear localization and activation [32]. In agreement with these findings, our results showed that partial IPO7 depletion increased the nuclear accumulation of RPL5 and RPL11 along with MDM2, which increased p53 stability (Figure 3A,B). In all experiments, RPL11 and RPL5 appeared to be highly expressed in the cytoplasm of PTC cells and in the nuclei and cytoplasm of normal thyroid cells, Nthy-ori 3-1 cells (Figure 3A,B; Figure 4C,D). It is worth mentioning here that RPL11 was found to downregulate the activity of E2F1, p53, and c-Myc [56,57], which support our results showing reduced activity of these pathways in PTC cells (Table 2). In summary, this work showed that high expression of IPO7 contributes to reduced p53 stability and activity through regulation of the RP-MDM2-p53 pathway. This IPO7-mediated mechanism can be considered an addition to the list of p53 posttranslational modifications that occur in tumorigenesis. 

Parallel experiments conducted in the normal thyroid cell line (Nthy-ori 3-1) showed that miR-7-5p loss of function, with the consequent IPO7 upregulation, promoted a p53 nucleocytoplasmic translocation (Figure 4A,B). However, the change in the p53 protein level upon modulation of IPO7 or miR-7-5p expression in the normal cell line did not reach statistical significance (Appendix A). Viability assay showed the expected positive effects on cell proliferation in the normal cells upon miR-7-5p loss of function, but with no statistical significance (Appendix A). The nucleocytoplasmic translocation of RPL11 and MDM2 in Nthy-ori 3-1 cells after inhibition of miR-7-5p was clear, unlike RPL5, which remained to be strongly detected in the nuclei and cytoplasm of the treated cells (Figure 4C,D). A possible explanation of these findings is that the miR-7-5p/IPO7-mediated regulation of p53 stability is more efficient in tumor cells where additional tumorigenesis regulatory mechanisms come into play. This might have an important implication in designing therapeutics to target the p53 pathway. The available therapies, such as MDM2 inhibitors, are designed to stabilize p53 in tumor cells. However, clinical data showed that they have a potential problem of stabilizing p53 in normal cells, which constitutes a major challenge to the success of these treatments [58]. Moreover, preclinical and clinical data showed that targeting importins constitutes a new therapeutic avenue in treating malignancies [59]. Therefore, modulation of miR-7-5p and IPO7 expression might be a potential tool in restoring p53 function specifically in cancer cells.

In conclusion, miR-7-5p and IPO7 have important roles in regulating the dynamics and stability of p53 in thyroid cells. In PTC, miR-7-5p downregulation with the consequent upregulation of IPO7 contributes to p53 suppression through modulating the nucleocytoplasmic trafficking of p53, MDM2, and ribosomal proteins (RPL5/RPL11). Deregulation of miR-7-5p and IPO7 might be a protective mechanism hijacked by cancer cells to evade p53 growth arrest in situations of stress associated with carcinogenesis. These results are important for designing therapies to restore normal p53 function in cancer cells.

## 4. Materials and Methods

### 4.1. Tissue Samples

Fresh and formalin-fixed paraffin-embedded (FFPE) thyroid tissues were obtained from histopathology units at the Kuwait Cancer Control and Farwaniya hospitals. All cases were microscopically reviewed by two consultant histopathologists following the WHO classification of endocrine tumors and as described in our previous published work [12,34,60]. The groups were classified as follows: 1. Classic PTC (cPTC) includes conventional infiltrative PTC composed predominantly of papillae. 2. NIFTPs are noninvasive follicular neoplasms with PTC nuclear features. 3. Follicular nodular disease (FND) is the benign growth previously referred to as multinodular goiter. Only samples with a definite classification that fits within these three groups were included in the study. The FFPE samples used in this study included 30 cPTCs, 11 NIFTPs, and 10 cases of follicular nodular disease (FND), which were used as controls. Fresh thyroid tissues were portions of thyroid specimens from partial or total thyroidectomy that were left over after routine gross and histopathological procedures, and they included 8 PTCs and 3 FND samples used as controls.

### 4.2. Thyroid Cell Culture and Transfection Experiments

Primary thyroid cells were isolated, characterized, and maintained in culture as described in our previously published work [60]. In all functional experiments, the primary cells were from early passages (Passages 1–3). Nthy-ori 3-1 normal human thyroid follicular epithelial cell line (ECACC, catalog # 90011609) was purchased from The European Collection of Authenticated Cell Cultures (ECACC) UK, and cultured according to manufacturer’s recommendations. Cells from passage 10 were used in the functional experiments. For transfection experiments, primary cells and cell lines were plated at a density of 1 × 10^5^ cells/well in antibiotic-free minimal essential media. Transfection was performed using 15 µL of HiPerFect Transfection Reagent (Qiagen, Hilden, Germany) and 0.5 ng of miScript miR-7-5p mimic (Qiagen, Gene globe ID-MSY0000252), or inhibitor (Qiagen, Gene globe ID-YI04100814-DDA), 10 nM IPO7 siRNA (Qiagen, Gene globe ID-SI00088081), or negative control (Qiagen). The cells were incubated for 48 h before RNA/protein extraction and subsequent experiments. The success of the transfection was confirmed by testing the expression of IPO7 and miR-7-5p in treated cells by real-time PCR.

### 4.3. Luciferase Reporter Assay

The cell death signaling pathways were tested using the Cignal Finder Cancer Pathway Reporter Array (Qiagen). This array tests for many pathways related to cell death and includes p53, pRb/E2F, Myc/Max, NFKB, and JNK. Transfected cells and control cells were plated at a density of 8 × 10^4^ cells/well in 96-well plates containing transcription factor-responsive reporters, negative controls, and positive controls. After 48 h of incubation, the firefly luciferase activity was stopped by adding 100 µL of Dual-Glo^®^ Stop & Glo^®^ Reagent (Promega, Madison, WI, USA). Then, 100 µL of Dual-Glo^®^ Luciferase Reagent (Promega, Madison, WI, USA) containing buffer and substrate was added, and the luminescence was measured by a Thermo Scientific Appliskan Plate Reader, Waltham, MA, USA. Firefly/Renilla activity ratios were generated for experimental and negative control transfections. The change in the activity of each signaling pathway was determined by comparing the normalized luciferase activities of the reporter in experimental versus control transfectants using the following formula: fold change = (firefly/Renilla ratio of experimental sample)/(firefly/Renilla ratio of control sample). All transfections were performed in quadruplicate for each of the reporter assays. Transfection efficiency was estimated by observing GFP expression (a constitutively expressing GFP construct containing Monster Green^®^ Fluorescent Protein, Promega, Madison, WI, USA) in the positive control wells by fluorescence microscopy. A transfection efficiency higher than 80% was considered acceptable.

### 4.4. Reverse Transcription and Quantitative Real-Time PCR Amplification

Total RNA from cultured cells and paraffin-embedded thyroid tissues was isolated using TRIzol (Thermo Fisher Scientific Inc, Waltham, MA, USA) and miRNeasy FFPE Kit (Qiagen, Hilden, Germany), respectively, following the manufacturers’ instructions. Reverse transcription of cDNA and RT-PCR were performed using miScript II RT Kit (Qiagen), RT^2^ SYBR Green Mastermix (Qiagen), and RT^2^ qPCR Primer Assay-specific primers (Qiagen), following standard procedures. HPRT was used as the housekeeping gene since it showed the most stable expression pattern among different thyroid lesions as well as different experimental conditions as per our previous published results [61]. PCRs were run on an ABI 7500 Fast Block real-time PCR machine. The cycling conditions included an initial holding stage at 95.0 °C for 10 min, followed by 40 amplification cycles consisting of denaturation at 95.0 °C for 15 s and annealing/extension at 60.0 °C for 1 min. All samples were run in triplicate, and the mean Cq value with standard deviation was calculated for each sample. The expression in the test groups was compared to that in the control group using the formula (ΔΔCq = ΔCq of test group − ΔCq of control group), and the results are presented as the fold change (2^−ΔΔCq^) [62].

### 4.5. Immunofluorescence and Confocal Microscopy

The expression and subcellular localization of different molecules in FFPE tissues and cultured cells were tested by indirect immunofluorescence staining and confocal microscopy following standard procedures. The FFPE tissues stained included 30 PTC, 10 NIFTP, and 10 FND cases. The antibodies used were anti-p53 (Cell Signaling, Danvers, MA, USA), anti-MDM2 (Novus Biologicals, Centennial, CO, USA), anti-RPL5 (Protein Tech, Rosemont, IL, USA), anti-RPL11 (Novus Biologicals, Centennial, CO, USA), and anti-IPO7 (Protein Tech, Rosemont, IL, USA). Secondary antibodies conjugated to Alexa Fluor 555, Alexa Fluor 488 (Invitrogen, Waltham, MA, USA) or HRP (Dako, Carpinteria, CA, USA) were used. Staining was assessed qualitatively and scored as positive or negative. A positive score was given only when more than 10% of the tumor follicular cells showed non-ambiguous staining. Positive staining in immune cells or other cells of the tumor microenvironment was not considered positive. The expression and cellular localization of miR-7-5p were studied by in situ hybridization using miRCURY LNA miRNA Detection Probes (QIAGEN). The protocol is described in our previous work [12]. LSM 700 and 800 laser scanning confocal microscopes (Zeiss, Oberkochen, Germany) and Zen software (version 14.0.0.201, Zeiss) were used for acquisition and analysis of images. Mean fluorescence intensity is provided in treated cells and controls. Colocalization metrics were computed using Zen software (Zeiss, version 14.0.0.201, Germany) based on the co-occurrence of the identified signals and their relative intensities. Mean nuclear localization intensity (NI) is calculated from the colocalization data (NI = the number of green/red pixels (positive for the target) that colocalize with DAPI nuclear stain (localized in the nucleus).

### 4.6. Immunoblotting

Protein extracted from treated cells or controls (20 μg) was mixed with loading buffer and rainbow marker (14,300–220,000 Da, Amersham Pharmacia Biotech Ltd., Amersham, UK). Electrophoresis was performed using SDS-PAGE (SDS-PAGE; 5–14% polyacrylamide gradient gels). Protein was transferred to nitrocellulose membranes at a stable current of 75A for 75 min at room temperature. Membranes were stained with Ponceau and used to estimate total proteins. Blocking was performed for 1 h at room temperature with 1× TBS 1% casein blocker (Bio-Rad, Hercules, CA, USA). The membranes were incubated with anti-p53 antibody (Cell Signaling) for one hour at room temperature followed by a secondary antibody. Protein bands were detected by chemiluminescence (ECL) and the quantity of p53 was normalized to the total protein using ChemiDoc MP Imaging System and Image Lab Software Version 6.1, 2017, Bio-Rad Laboratories, Inc.

### 4.7. Viability Assay

Cultured cells were harvested by trypsinization and then centrifuged and washed with cold PBS. Viable cells were counted by Trypan Blue Exclusion Test using Beckman Coulter VI-Cell XR. For the apoptosis assay, the cells were resuspended in 400 µL of binding buffer (Invitrogen). Two microliters of both Annexin V and propidium iodide were added, and the cells were incubated on ice for 20 min. The stained cells were analyzed by flow cytometry following standard procedures.

### 4.8. Statistical Analysis

The gene expression levels in different groups were compared using Student’s *t* test. The difference in gene and protein expression levels between matched samples (transfected with IPO7 siRNA or miR-7-5p mimic/inhibitor versus control) was determined using paired-sample *t* tests. Statistical tests were chosen after confirming the normal distribution of the obtained data. In all tests, *p*-values < 0.05 were considered to indicate statistical significance.

## Figures and Tables

**Figure 1 ijms-26-05813-f001:**
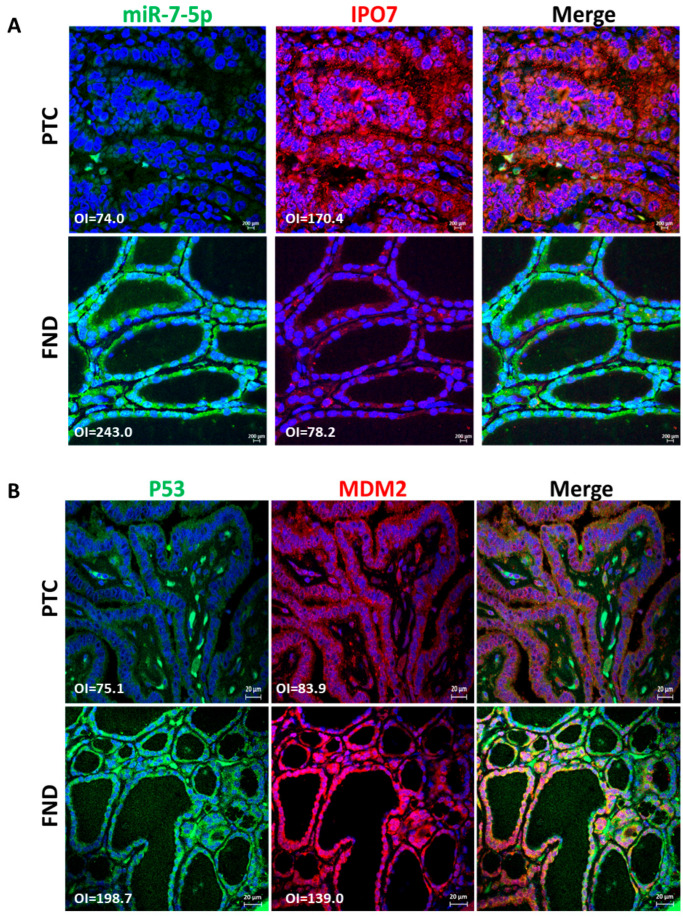
Immunofluorescence staining images showing the protein expression of IPO7 and p53 in representative PTC and FND tissues. (**A**) IPO7 (red) is localized in the cytoplasm and nuclei of thyroid follicular cells in PTC samples, while it is expressed at low intensity in FND. MiR-7-5p (green) tested by in situ hybridization is negative in PTC tumor cells and strongly positive in FND. (**B**) Tumor cells in PTC show negative expression of p53 (green) and low cytoplasmic/nuclear expression of MDM2 (red). P53 staining was detected only in infiltrating cells in PTC tissues. In FND tissue, p53 and MDM2 showed strong cytoplasmic and nuclear expression. Nuclei were counterstained with DAPI (blue). The scale bar is shown in the bottom right panel = 20 μm. The mean fluorescence intensity was analyzed by Zen software (version 14.0.0.201, Zeiss) and presented in the pictures as overall intensity (OI).

**Figure 2 ijms-26-05813-f002:**
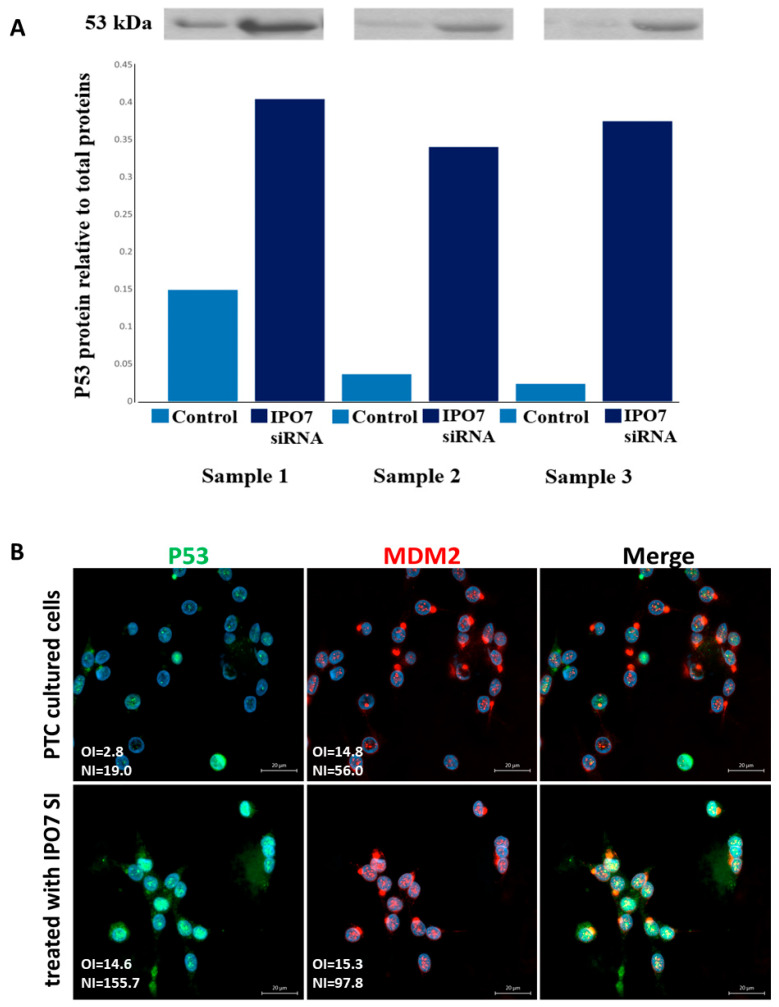
Partial depletion of IPO7 stabilizes p53. (**A**) Immunoblots showing the expression of p53 in PTC cells transfected with IPO7 siRNA or the control. The bar graph shows the quantity of p53 protein measured relative to total protein. The results of three samples are shown. A significant increase in p53 protein was detected in cells treated with IPO7 siRNA (*p* = 0.0026). (**B**) Immunofluorescence images showing that p53 (green) and MDM2 (red) shift to the nuclei of PTC cells after partial IPO7 depletion (*p* = 0.0006 and *p* = 0.005, respectively). Three PTC samples were tested and images from a representative sample are shown. Nuclei were counterstained with DAPI (blue). Images were analyzed by LSM 800 and Zen software (Zeiss). The mean fluorescence intensity is presented in the pictures as overall intensity (OI). The mean nuclear localization intensity is calculated from the colocalization data. Nuclear intensity (NI) = the number of green/red pixels that colocalize with DAPI nuclear stain. The increased NI score in the treated cells indicates increased nuclear expression. NI scores across samples were analyzed using paired samples *t* test. The scale bar is shown in the bottom right panel = 20 μm.

**Figure 3 ijms-26-05813-f003:**
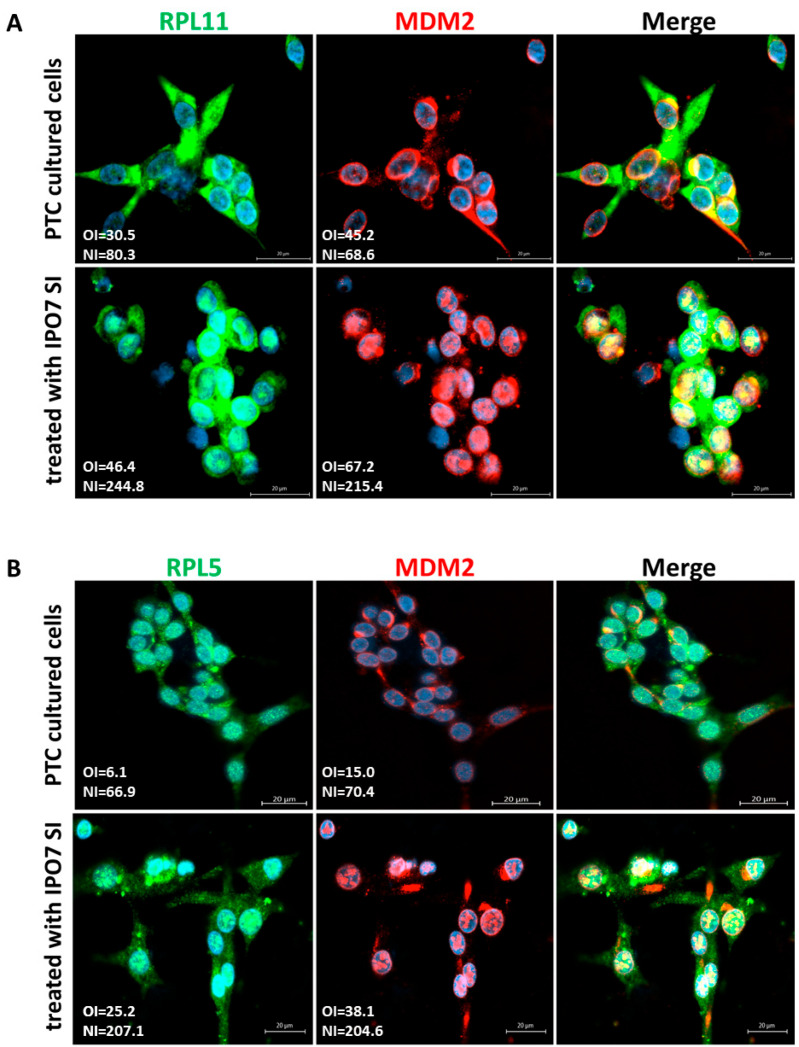
Effect of partial IPO7 depletion on the cellular localization of ribosomal proteins in primary cultured PTC cells. (**A**) RPL11 (green) shifts from the cytoplasm to the nuclei of PTC cells in colocalization with MDM2 (red) after partial IPO7 depletion (*p* < 0.001 and *p* = 0.002, respectively). (**B**) RPL5 (green) and MDM2 (red) colocalize in the nuclei of PTC cells after partial IPO7 depletion (*p* = 0.009 and *p* = 0.02, respectively). Three PTC samples were tested and images from a representative sample are shown. Nuclei were counterstained with DAPI (blue). Images were analyzed by LSM 800 and Zen software (Zeiss). The mean fluorescence intensity is presented in the pictures as overall intensity (OI). The mean nuclear localization intensity is calculated from the colocalization data. Nuclear intensity (NI) = the number of green/red pixels that colocalize with DAPI nuclear stain. The increased NI score in the treated cells indicates increased nuclear expression. NI scores across samples were analyzed using paired samples *t* test. The yellow color in the merged pictures indicates the colocalization of the red and green fluorescence. The scale bar is shown in the bottom right panel = 20 μm.

**Figure 4 ijms-26-05813-f004:**
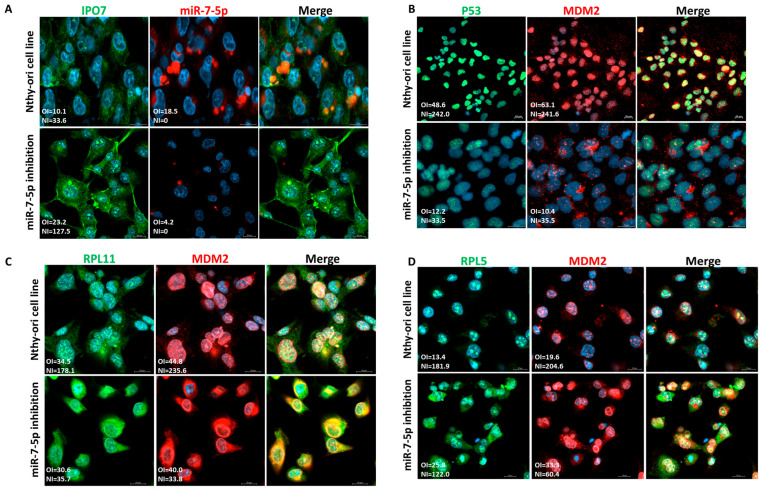
Functional experiments in the normal thyroid cell line (Nthy-ori 3-1). Each experiment was repeated three times and results from one representative experiment are shown. (**A**) Nthy-ori 3-1 cells express low levels of IPO7 (green) and moderate levels of miR-7-5p (red). Transfection with miR-7-5p inhibitor increased the expression of IPO7 in the treated cells. (**B**) Nthy-ori 3-1 cells show nuclear expression of p53 (green) and MDM2 (red). P53 and MDM2 shifted to the cytoplasm in cells transfected with the miR-7-5p inhibitor (*p* = 0.01 and *p* < 0.001, respectively). (**C**) RPL11 and MDM2 are located in the cells’ nuclei. MiR-7-5p loss of function promoted the nucleocytoplasmic translocation of RPL11 and MDM2 as indicated by the reduced NI (nuclear intensity) score of the treated cells (*p* = 0.002 and *p* < 0.001, respectively). (**D**) RPL5 and MDM2 are located in the nuclei of cells. MiR-7-5p loss of function promoted the nucleocytoplasmic translocation of RPL5 and MDM2 (*p* = 0.06, *p* = 0.003). RPL5 retained an intense staining in the cytoplasm and nuclei of the treated cells. Nuclei were counterstained with DAPI (blue). Images were analyzed by LSM 800 and Zen software (Zeiss). The mean fluorescence intensity is presented in the pictures as overall intensity (OI). The mean nuclear localization intensity is calculated as nuclear intensity (NI) = the number of green/red pixels that colocalize with DAPI nuclear stain. NI scores across samples were analyzed using paired samples *t* test. The yellow color in the merged pictures indicates the colocalization of the red and green fluorescence. The scale bar is shown in the bottom right panel = 20 μm.

**Table 1 ijms-26-05813-t001:** Expression of IPO7 and miR-7-5p in different thyroid tissue samples.

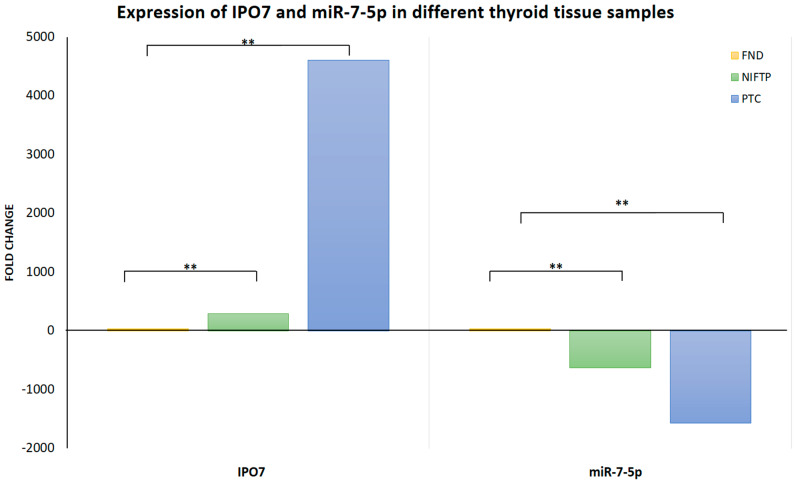							
	**Mean Expression Value** **ΔCq (SD)**			**NIFTP vs. FND**		**PTC vs. FND**	
	**FND** **(*n* = 10)**	**NIFTP** **(*n* = 11)**	**PTC** **(*n* = 30)**	**Fold Change**	***p*-Value ***	**Fold Change**	***p*-Value ***
**IPO7**	2.35 (1.48)	−5.83 (3.25)	−9.82 (3.86)	290.02	<0.0001	4608.24	<0.0001
**miR-7-5p**	−1.05 (1.15)	8.25 (5.07)	9.57 (4.17)	−630.35	0.0003	−1573.76	<0.0001

Fold change is calculated using the 2^−(ΔΔCq)^ formula. * Statistical analysis was performed using Student’s *t* test. ** indicates statistical significance of <0.001.

**Table 2 ijms-26-05813-t002:** Endogenous activity of cell death signaling pathways in PTC cultured cells measured by luciferase assay.

	miR-7 Level ^a^	Signaling Pathways ^b^				
		p53	pRb/E2F	MYC/MAX	NFκB	MAPK/JNK
Sample 1	−22.85	−6.59	−2.67	−4.51	−1.88	−11.38
Sample 2	−15.37	−3.70	−3.64	−3.76	−1.46	−58.01
Sample 3	−31.71	1.47	−1.96	−1.90	−1.14	−152.93
Sample 4	−2.38	−4.63	−5.87	−7.41	−2.45	−15.54
Sample 5	−62.77	−3.14	−1.97	−4.26	−1.23	−14.69
Sample 6	−13.97	−2.26	1.99	1.35	4.09	−1.28
Sample 7	−43.74	−1.78	−2.39	−2.08	1.29	−5.97
Sample 8	−17.72	−2.45	−1.93	−1.71	1.63	−3.29
Mean (SD) *p*-value	−26.3 (19.2) 0.006	−2.89 (2.34) 0.01	−2.3 (2.19) 0.02	−3.04 (2.58) 0.012	−0.14 (2.25) 0.86	−32.89 (51.74) 0.11

a: miRNA level measured by RT-PCR and expressed in fold change in PTC compared to FND controls; b: Activity of cell death signaling pathways expressed as fold change of the luciferase activity in PTC (primary cultured cells from 8 different tumor samples) compared to FND (primary cultured cells from 3 different samples). Statistical analysis is performed using Student’s *t* test.

**Table 3 ijms-26-05813-t003:** Effect of miR-7-5p gain of function on IPO7 expression and cell death signaling pathway activity in PTC.

Sample (Diagnosis)	Expression Level ^a^		Signaling Pathway Activity ^b^				
			p53	pRb/E2F	Myc/Max	NFκB	MAPK/JNK
	miR-7-5p	IPO7	p53	E2F/DP1	Myc/Max	NFκB	AP-1
1 (PTC)	22.50	−92.72	8.23	4.86	6.01	1.43	1.62
2 (PTC)	43.47	−19.41	5.29	−1.22	1.71	1.87	−1.42
3 (PTC)	48.23	−7.45	2.72	1.32	4.34	1.73	1.28
4 (PTC)	201.11	−16.87	1.07	1.24	1.14	1.64	1.70
5 (PTC)	36.05	−12.27	2.60	2.86	1.84	−1.36	−1.86
*p*-Value	<0.002		0.04	0.067	0.01	0.45	0.38

IPO7 is a predicted target of miR-7-5p with a complementary sequence at position 1909–1916 of its 3′ untranslated region [TargetScan Database]. a: Expression level calculated as the fold change of miR-7-5p and IPO7 in primary cultured PTC cells transfected with miR-7-5p mimic compared to cells transfected with a negative control; fold change is calculated using the 2^−(ΔΔCq)^ formula. b: Pathway activity expressed as the fold change of the luciferase activity of the same primary cultured PTC cells transfected with miR-7-5p mimic compared to cells transfected with a negative control. Results from experiments performed on five different PTC cases are shown. Statistical analysis is performed using paired Student’s *t* test.

## Data Availability

The datasets generated during and/or analyzed during the current study are available from the corresponding author upon reasonable request.

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
