# Peer review of "miR-7-5p and Importin-7 Regulate the p53 Dynamics and Stability in Malignant and Benign Thyroid Cells"

_ijms, 2025, doi:10.3390/ijms26125813_

Round 1
Reviewer 1 Report
Comments and Suggestions for Authors
Peer-Review
miR-7-5p and Importin-7 Regulate the p53 Dynamics and Stability in Malignant and Benign Thyroid Cells
Title
Comment: Please, include the type of study in the title (e.g., "An Experimental Study" or "An In Vitro and Ex Vivo Functional Study").
Abstract
Comment: In the Abstract, the authors omit a clear statement of the study objective. Moreover, providing selected numerical results (such as fold-changes or p-values) would substantiate the conclusions and make the Abstract more engaging for readers.
Introduction
Comment: The last paragraph of the introduction describes a result rather than stating a clear research objective. It would be more appropriate to rephrase this section to present the study's objective in a prospective manner, avoiding anticipation of the findings. (e.g., "This study aims to investigate whether...").
Methods
Comment: 2.1 (Tissue Samples) While the description of tissue sources and histopathological classification is appropriate, the Methods section would benefit from clearly stating the inclusion and exclusion criteria for sample selection.
Comment: 2.3 (Luciferase Reporter Assay) The luciferase reporter assay methodology is appropriate and well-controlled. However, it would improve clarity if the authors explicitly list the transcription factors or signaling pathways assessed by the Cignal Finder Reporter Array in the Methods section.
Comment: 2.4 (RT-qPCR) It is recommended that the authors align the RT-qPCR description with the MIQE (Minimum Information for Publication of Quantitative Real-Time PCR Experiments) guidelines. Specifically, the use of “Cq” (quantification cycle) instead of “Ct” (cycle threshold) is advised, as “Cq” is the standardized and currently preferred terminology to ensure consistency across publications.
Comment: 2.4 (RT-qPCR) Although the authors cite previous publications (references 36 and 37) for the primer sequences, it is recommended to explicitly provide the specific primers used, either in the Methods section or as supplementary material, to ensure reproducibility. According to the MIQE guidelines, it is also important to report the amplification efficiencies of both target and reference genes when using the ΔΔCq method, as these efficiencies must be comparable for accurate relative quantification.
Comment: 2.4 (RT-qPCR), the authors appropriately describe the reagents, equipment (ABI 7500 Fast Block real-time PCR machine), and calculation method. However, it is recommended to also include the thermal cycling conditions (temperatures, times, number of cycles, and melting curve analysis), following MIQE guidelines.
Comment: 2.8. (Statistical Analysis)Iit is recommended that the authors clarify whether they tested for the normality of data distribution and homogeneity of variances before applying parametric tests (t-test and ANOVA), as required to validate these analyses. If the data do not follow a normal distribution, the use of appropriate non-parametric statistical methods should be considered.
Results
Comment: 3.1. (Expression of IPO7 and p53 in PTC Clinical Samples ) it would strengthen the Rt-qPCR results presentation if the authors include selected key quantitative values (e.g., fold-changes and p-values) in the text alongside references to tables and figures, to better support the narrative and facilitate reader interpretation.
Comment: 3.2 (p53 Activity Is Suppressed in PTC and Is Rescued by miR-7-5p Gain of Function and more broadly throughout the Results section), it would improve the clarity and accessibility of the findings if the authors included selected key fold-change values and p-values directly within the text, rather than relying exclusively on tables and figures. This approach would better support the narrative and facilitate the reader's interpretation of the results.
Comment: 3.3 and 3.4, the authors could consider strengthening the colocalization analysis by incorporating quantitative metrics, such as Pearson’s correlation coefficient (PCC) or Manders’ overlap coefficient (MOC), similarly to fluorescence microscopy studies (Zinchuk et al., 2007, PMID: 17898874; Bolte & Cordelières, 2006, PMID: 17210054; McDonald & Dunn, 2013, PMC4428547).
Discussion
Comment: While the Discussion appropriately covers most key findings, the authors do not address two relevant results: (i) the lack of significant differences in p53 mRNA levels among groups (reported in Section 3.1), and (ii) the increase in miR-7-5p expression following IPO7 silencing (reported in Section 3.3). These results are relevant and should be considered for discussion. If these points have already been addressed, please indicate where, as they may have been inadvertently overlooked during the review.
Conclusion
Comment: The conclusions are largely supported by the presented results. However, could the authors indicate which experiments support the following statements, such as (i) "facilitates the engagement of RPL11/RPL5 in the ribosomal biogenesis stress response" and (ii) "promotes MDM2/p53 cytoplasmic repositioning and p53 degradation," to clearly clarify that these conclusions are supported by experimental data?
Author Response
Response to reviewer's comments
Thank you for reviewing my manuscript. My detailed response to your comments is found below with the corresponding corrections in track changes in the resubmitted file "revised manuscript".
Comment 1: Please, include the type of study in the title (e.g., "An Experimental Study" or "An In Vitro and Ex Vivo Functional Study").
Response: Thank you for the comment. I understand the validity of your point. However, my study includes descriptive data of experiments performed in clinical samples, as well as in vitro and ex vivo functional experiments. I found it difficult to reflect this in the title, therefore, I prefer to keep the title as it is.
Comment 2: In the Abstract, the authors omit a clear statement of the study objective. Moreover, providing selected numerical results (such as fold-changes or p-values) would substantiate the conclusions and make the Abstract more engaging for readers.
Response: I agree with the comment. A clear statement explaining the aim of the study is added to the abstract.
Comment 3: The last paragraph of the introduction describes a result rather than stating a clear research objective. It would be more appropriate to rephrase this section to present the study's objective in a prospective manner, avoiding anticipation of the findings. (e.g., "This study aims to investigate whether...").
Response: I agree with the comment. A sentence stating the objective of this study is added at the end of the introduction.
Comment 4: 2.1 (Tissue Samples) While the description of tissue sources and histopathological classification is appropriate, the Methods section would benefit from clearly stating the inclusion and exclusion criteria for sample selection.
Response: I agree with the comment. A sentence describing the clear inclusion criteria is added to the section on tissue samples in the Methodology.
Comment 5: 2.3 (Luciferase Reporter Assay) The luciferase reporter assay methodology is appropriate and well-controlled. However, it would improve clarity if the authors explicitly list the transcription factors or signaling pathways assessed by the Cignal Finder Reporter Array in the Methods section.
Response: I agree with the comment. Required details are added to the section on luciferase assay in the Methodology.
Comment 6: 2.4 (RT-qPCR) It is recommended that the authors align the RT-qPCR description with the MIQE (Minimum Information for Publication of Quantitative Real-Time PCR Experiments) guidelines. Specifically, the use of “Cq” (quantification cycle) instead of “Ct” (cycle threshold) is advised, as “Cq” is the standardized and currently preferred terminology to ensure consistency across publications.
Response: I agree with the comment. "Ct" is changed to "Cq' in the manuscript and in the tables.
Comment 7: 2.4 (RT-qPCR) Although the authors cite previous publications (references 36 and 37) for the primer sequences, it is recommended to explicitly provide the specific primers used, either in the Methods section or as supplementary material, to ensure reproducibility. According to the MIQE guidelines, it is also important to report the amplification efficiencies of both target and reference genes when using the ΔΔCq method, as these efficiencies must be comparable for accurate relative quantification.
Response: Thank you for this important comment. We acknowledge the recommendation to provide specific primer sequences to enhance reproducibility in line with MIQE guidelines. However, in this study, we used commercially available predesigned RT-qPCR primer assays from Qiagen. Qiagen does not disclose the specific primer sequences due to proprietary reasons. The target and the reference genes primers were also pre-validated by the manufacturer for optimal and comparable amplification efficiency.
Comment 8: 2.4 (RT-qPCR), the authors appropriately describe the reagents, equipment (ABI 7500 Fast Block real-time PCR machine), and calculation method. However, it is recommended to also include the thermal cycling conditions (temperatures, times, number of cycles, and melting curve analysis), following MIQE guidelines.
Response: I agree with the comment. Details about cycling conditions were added to the section on RT-PCR in the Methodology.
Comment 9: 2.8. (Statistical Analysis)Iit is recommended that the authors clarify whether they tested for the normality of data distribution and homogeneity of variances before applying parametric tests (t-test and ANOVA), as required to validate these analyses. If the data do not follow a normal distribution, the use of appropriate non-parametric statistical methods should be considered.
Response: I agree with the comment. I confirm that the selection of statistical tests was based on the distribution of the data. Only t-tests and paired t-tests are used in this study. The obtained data met the assumptions of normality and equal variances. A sentence clarifying this was added to the section on statistical analysis.
Comment 10: 3.1. (Expression of IPO7 and p53 in PTC Clinical Samples ) it would strengthen the Rt-qPCR results presentation if the authors include selected key quantitative values (e.g., fold-changes and p-values) in the text alongside references to tables and figures, to better support the narrative and facilitate reader interpretation.
Response: I agree with the comment. Values including fold changes and p values were added within the text in the results section.
Comment 11: 3.2 (p53 Activity Is Suppressed in PTC and Is Rescued by miR-7-5p Gain of Function and more broadly throughout the Results section), it would improve the clarity and accessibility of the findings if the authors included selected key fold-change values and p-values directly within the text, rather than relying exclusively on tables and figures. This approach would better support the narrative and facilitate the reader's interpretation of the results.
Response: I agree with the comment. Values including fold changes and p values were added within the text in the results section.
Comment 12: 3.3 and 3.4, the authors could consider strengthening the colocalization analysis by incorporating quantitative metrics, such as Pearson’s correlation coefficient (PCC) or Manders’ overlap coefficient (MOC), similarly to fluorescence microscopy studies (Zinchuk et al., 2007, PMID: 17898874; Bolte & Cordelières, 2006, PMID: 17210054; McDonald & Dunn, 2013, PMC4428547).
Response: Thank you for the comment. I understand the importance of calculating the colocalization coefficients. However, Pearson’s correlation coefficient (PCC) is mainly used to evaluate the pixel linear relationship between fluorophore intensities. Manders’ overlap coefficient (MOC) is used to assess the degree of signal overlap. Since we are more interested in the subcellular localization of our molecules (p53, MDM2, RPLs) and their nuclear expression across different experimental conditions/treatments, we used special quantitative colocalization metrics to complement our interpretation. This is explained in the methods section and the corresponding figures legends. "Colocalization metrics were computed using Zen software (Zeiss, version 14.0.0.201, Germany) based on the co-occurrence of the identified signals and their relative intensities. Mean nuclear localization intensity (NI) is calculated from the colocalization data (NI= the number of green/red pixels (positive for the target) that colocalize with DAPI nuclear stain (localized in the nucleus)". The increased NI score in the treated cells indicates increased nuclear expression.
Comment 13: While the Discussion appropriately covers most key findings, the authors do not address two relevant results: (i) the lack of significant differences in p53 mRNA levels among groups (reported in Section 3.1), and (ii) the increase in miR-7-5p expression following IPO7 silencing (reported in Section 3.3). These results are relevant and should be considered for discussion. If these points have already been addressed, please indicate where, as they may have been inadvertently overlooked during the review.
Response: I agree with the comment. Discussion of these two points was added to the second paragraph of the discussion (pages 12 and 13).
Comment 14: The conclusions are largely supported by the presented results. However, could the authors indicate which experiments support the following statements, such as (i) "facilitates the engagement of RPL11/RPL5 in the ribosomal biogenesis stress response" and (ii) "promotes MDM2/p53 cytoplasmic repositioning and p53 degradation," to clearly clarify that these conclusions are supported by experimental data?
Response: I agree with the comment. The conclusion has been changed to include only the statements supported by the experimental results of this study.
Reviewer 2 Report
Comments and Suggestions for Authors
Overall, this manuscript presents a well-conducted and clearly articulated study investigating the roles of miR-7-5p and Importin-7 (IPO7) in regulating p53 in papillary thyroid carcinoma (PTC). The authors combine analysis of clinical samples with functional experiments in primary cells and a normal thyroid cell line, showing that downregulation of miR-7-5p leads to IPO7 overexpression in PTC, which in turn contributes to p53 suppression by modulating the nucleocytoplasmic trafficking of p53, MDM2, and ribosomal proteins (RPL5/RPL11). The authors provide novel insights into p53 regulation in PTC and propose a potential therapeutic target.
The overall quality of the manuscript is high and only minor revisions are needed. Firstly, the authors should consider briefly mentioning the specific test used for comparisons directly in the figure legends (e.g., "Student's t-test, *p < 0.05") for figures like Figure 1, 2, 3, and 4. Secondly, despite the adoption of HPRT as the housekeeping gene is acceptable, providing a brief justification for its choice or confirming its stable expression across the conditions tested could be beneficial. Thirdly, I would also suggest the authors to expand the introduction section providing more information about PTC, its molecular background, diagnosis and treatment modalities (doi: 10.1158/1055-9965.EPI-21-1440; doi: 10.1530/ETJ-22-0146; 10.3389/fendo.2023.1101410; https://doi.org/10.1002/cncy.22224; 10.3390/diagnostics11061043; doi: 10.1002/cncy.22120). Lastly, a moderate English editing and spell and punctuation check should be performed.
Author Response
Response to reviewer's comments
Thank you for reviewing my manuscript. My detailed response to your comments is found below with the corresponding corrections in track changes in the resubmitted file "revised manuscript".
Comment 1: Overall, this manuscript presents a well-conducted and clearly articulated study investigating the roles of miR-7-5p and Importin-7 (IPO7) in regulating p53 in papillary thyroid carcinoma (PTC). The authors combine analysis of clinical samples with functional experiments in primary cells and a normal thyroid cell line, showing that downregulation of miR-7-5p leads to IPO7 overexpression in PTC, which in turn contributes to p53 suppression by modulating the nucleocytoplasmic trafficking of p53, MDM2, and ribosomal proteins (RPL5/RPL11). The authors provide novel insights into p53 regulation in PTC and propose a potential therapeutic target. The overall quality of the manuscript is high and only minor revisions are needed.
Response: Thank you for your valuable positive comments.
Comment 2: The authors should consider briefly mentioning the specific test used for comparisons directly in the figure legends (e.g., "Student's t-test, *p < 0.05") for figures like Figure 1, 2, 3, and 4.
Response: I agree with this important comment. The statistical significance of the nuclear colocalization intensity scores compared across different experiments (p values) was added to the legends in Figures 2, 3 and 4.
Comment 3: Despite the adoption of HPRT as the housekeeping gene is acceptable, providing a brief justification for its choice or confirming its stable expression across the conditions tested could be beneficial.
Response: I agree with this important comment. A statement was added in paragraph #5 in the methodology section. It included the justification of using HPRT as house-keeping gene based on our previously published paper. The new reference is added to the references list (reference #38).
Comment 4: I would also suggest the authors to expand the introduction section providing more information about PTC, its molecular background, diagnosis and treatment modalities (doi: 10.1158/1055-9965.EPI-21-1440; doi: 10.1530/ETJ-22-0146; 10.3389/fendo.2023.1101410; https://doi.org/10.1002/cncy.22224; 10.3390/diagnostics11061043; doi: 10.1002/cncy.22120).
Response: Thank you for the comment. I have added information on molecular alterations commonly involved in PTC (First paragraph of the introduction, page 3).
Comment 5: A moderate English editing and spell and punctuation check should be performed.
Response: An English language check has been performed.
Round 2
Reviewer 1 Report
Comments and Suggestions for Authors
The authors have addressed all my points and can now be accepted.